# Guidelines for Selecting Interlayer Spacers in Synthetic 2D-Based Antiferromagnets from First-Principles Simulations

**DOI:** 10.3390/nano9121764

**Published:** 2019-12-11

**Authors:** Ramón Cuadrado, Miguel Pruneda

**Affiliations:** 1Catalan Institute of Nanoscience and Nanotechnology - ICN2, CSIC and BIST, Campus UAB, 08193 Bellaterra, Spain; 2Universitat Autonoma de Barcelona, 08193 Bellaterra (Cerdanyola del Valles), Spain

**Keywords:** magnetic materials, magnetic anisotropy, exchange interactions, density functional theory, 2d materials, electronic structure

## Abstract

Following the recent synthesis of graphene–based antiferromagnetic ultrathin heterostructures made of Co and Fe, we analyse the effect of the spacer between both ferromagnetic materials. Using density functional calculations, we carried out an exhaustive study of the geometric, electronic and magnetic properties for intercalated single Co MLs on top of Ir(111) coupled to monolayered Fe through *n* graphene layers (*n* = 1, 2, 3) or monolayered *h*-BN. Different local atomic arrangements have been considered to model the Moiré patterns expected in these heterostructures. The magnetic exchange interactions between both ferromagnets ($J_{Co-Fe}$) are computed from explicit calculations of parallel and anti-parallel Fe/Co inter–layer alignments, and discussed in the context of recent experiments. Our analysis confirms that the robust antiferromagnetic superexchange–coupling between Fe and Co layers is mediated by the graphene spacer through the hybridization of C’s $p_{z}$ orbitals with Fe and Co’s 3*d* states. The hybridization is substantially suppressed for multilayered graphene spacers, for which the magnetic coupling between ferromagnets is critically reduced, suggesting the need for ultrathin (monolayer) spacers in the design of synthetic graphene-based antiferromagnets. In the case of *h*–BN, $p_{z}$ orbitals also mediate *d*(Fe/Co) coupling. However, there is a larger contribution of local ferromagnetic interactions. Magnetic anisotropy energies were also calculated using a fully relativistic description, and show out–of–plane easy axis for all the configurations, with remarkable net values in the range from 1 to 4 meV.

## 1. Introduction

In recent years, antiferromagnetic (AFM) materials have made an appearance in the field of spintronics, where are now considered as components for key active elements, rather than playing a mere supporting role. In addition to their potential for reducing device power consumption, stabilizing the magnetization of nanoscale particles at room temperature, and insensitivity to magnetic fields, they show large magnetotransport effects and ultrafast dynamics which can reach the THz range [1,2]. Among the possible materials that exhibit antiferromagnetism, there is a growing interest in the so called synthetic antiferromagnets (SAF), which can be engineered by stacking thin films of magnetic and non-magnetic materials. The technology has been developed since the late 1980s, but the apperance of 2D materials, mostly graphene but also *h*-BN and dichalcogenides, has opened new exciting opportunities in spintronics, for example in magnetic tunnel junctions (MTJ) [3].

The coupling between the two magnetic layers in SAF is typically determined by the nature of the non-magnetic spacer, and can be described via a Ruderman-Kittel-Kasuya-Yosida (RKKY) interaction [4]. This coupling is weaker than the direct exchange or superexchange coupling in AFM crystals. However, it is easier to manipulate. For metallic spacers, there is an oscillatory behaviour in the interlayer exchange coupling as a function of the spacer thickness, *z*. By changing the thickness one can tune the interaction from FM to AFM, with a period of a few Angstrom. On top of the oscillations, there a $\frac{1}{z^{2}}$ decay that suppresses the coupling for thick spacers [4]. For semiconducting spacers, the exchange coupling is frequently AFM, but it decays exponentially with a decay length of a couple of Angstroms. Semimetallic spacers, on the other hand, have a rather different behaviour, with AFM coupling that have longer decay lengths of up to tens of Angstroms [5,6].

Recently, a strong exchange coupling above room temperature has been reported in graphene-based synthetic antiferromagnets [7]. In these heterostructures, a Co monolayer (ML) is intercalated under graphene deposited on top of Ir(111), and subsequently covered by a Fe overlayer. In agreement with previous works, graphene seems to promote large perpendicular magnetic anisotropy (PMA) at the interface with the magnetic thin films [8]. First-principles calculations also confirm the direct role that graphene has on sustaining the antiferromagnetic superexchange-coupling between Fe and Co [7]. The structure of these SAF is very similar to that of MTJ where graphene acts as barrier material.

Integration of 2D materials in MTJ has advanced significantly in the last few years. Since the early work by Karpan [9], special attention has been given to spin filtering properties, which can lead to very high tunneling magnetoresistances (TMR). The dependence of the spin filtering, and magnetoresistance, with different ferromagnetic (FM) materials, and multilayered graphene thicknesses have been theoretically predicted [9,10,11]. The increase in the spin filtering effect with the number of layers has been confirmed experimentally [12,13], and the hybridization between the π states of graphene’s first layer and the ferromagnet established to play a critical role in the spin polarization sign [14]. Other 2D materials such as the insulating *h*-BN [15,16] or semiconducting transition metal dichalcogenides have been considered as ultrathin tunnel barriers in spin valves.

From the experimental point of view, these structures are very sensitive to the detailed atomic structure and chemical composition of the interface and knowledge of the structure is mandatory to analyze spin–filtering behaviour. First principles calculations give an idealised theoretical insight of the structural, electronic and magnetic properties of the MTJ based on 2D materials. The strength of the interlayer exchange coupling was predicted to be rather high when single layers of graphene or *h*-BN are used as spacers, and the sign of the interation is sensitive to the transition metal considered (AFM for Fe or Co, FM for Ni) [11]. However, although the 2D materials offer sharp interfaces, and layer–by–layer control of the thickness, in many cases the lattice constant mismatch between the ferromagnet and the honeycomb lattice of the spacer results in long-wavelength Moiré patterns [17,18], that are difficult to model from first principles [19,20]. The Moiré superlattice means that there is a variety of *local* stackings for the interfacial atoms, and each of them can give different magnetic interactions, modifying the picture that model calculations can give.

The objective of this manuscript is to present some guidelines for the design of new synthetic magnets based on 2D materials, advancing from previous theoretical works and following recent experimental developments in the literature. First, we extend a previous study where we established that ultra–thin synthetic Ir(111)/Co/Gr/Fe/ films exhibit robust perpendicular AFM superexchange–coupling between Fe and Co magnetic MLs [7]. Such coupling takes place throughout direct interfacial hybridisation between (Fe/Co)–3$d_{z^{2}}$ and C–$p_{z}$ states. Now we analyze in detail the influence that the optimized geometries have on the electronic and magnetic behaviour, and the effect of a possible increase in the thickness of the spacer (number of graphene layers of up to three). Furthermore, we also consider replacing semimetalic graphene by insulating *h*-BN, an ultimate thickness spacer for magnetoresistive junctions, with large magnetoresistance ratio and strong interlayer exchange couplings [11]. This provides more information on which orbitals are involved in the magnetic coupling and the influence of B and N species.

The paper is structured as follows: Section 2 describes the theoretical tools used in our study. Next, Section 3 and Section 4 include the results for Ir(111)/Co/*n*Gr/Fe and Ir(111)/Co/*h*-BN/Fe, respectively. Each one of these sections is split in subsections where we describe: the atomic structural geometries and energetic analysis (A), the electronic structure (B), and the magnetic anisotropy energy (C). Finally, conclusions are outlined in Section 5.

## 2. Methodology

To investigate the nature of the coupling between FM materials in graphene (*h*-BN)–based AFM structures, we perform first principles calculations based on DFT, and analyze the role played by semimetallic (single and multilayered graphene) or insulating (monolayered *h*-BN) spacers, using the same ferromagnets: Fe and Co MLs. We specifically perform geometrical, electronic and magnetic study of Co/*n*Gr/Fe [n=1,2,3] and Co/*h*-BN/Fe ultra–thin films stacked on Ir(111) using SIESTA code [21]. Our calculations are based on the generalized gradient approximation (GGA) for exchange-correlation (XC) following the Perdew, Burke, and Ernzerhof (PBE) version [22]. To describe the core electrons we use norm–conserving pseudopotentials in their fully separable Kleinmann–Bylander [23] form under the Troullier–Martins parametrisation [24]. To improve the description of the magnetic systems, pseudocore corrections are used to include in the XC terms not only the valence charge density, but also the core charge as Louie et al. pointed out [25]. We use double-ζ polarized (DZP) with strictly localized numerical atomic orbitals as basis set, and the electronic temperature –kT in the Fermi–Dirac distribution—is set to 5 meV. We ensure total energy convergence by considering a *k* sampling defined by a (21 × 21) grid, i.e., 441 *k* points within the Brillouin Zone (BZ) integration. Real–space integrals are computed over a three–dimensional grid with a resolution of 1000 Ry, a mesh fine enough to ensure convergence of the magnetic properties. A vacuum region of ~30.0Å is taken to avoid interaction between periodic replicas of the slab. Self-consistent (SC) convergence tolerance for the density matrix is set to 10^−6^ eV.

A two–dimensional periodic slab is built from an Ir(111) substrate, an intercalated Co ML under graphene or *h*-BN, and an extra Fe ML deposited on top. Although we take symmetric slabs – built as Fe/Gr/Co/9 × Ir(111)/Co/Gr/Fe– in the following we will refer only to one half of the structure as Ir(111)/Co/Gr/Fe. Due to the lattice mismatch between graphene and Ir(111), a Moiré pattern is expected in the hetero-structure (the Co layer can be considered pseudomorphic with the Ir lattice), with a superlattice wavelength of 2.5 nm [17]. Previous simulations have considered the whole Moiré cell for cobalt-intercalated graphene [19,20], although these studies do not take into account spin-orbit effects in the magnetic interactions. Rather than simulating this superstructure (with the 10 × 10 graphene unit cell and 9 × 9 Co/Ir lattice, which remains a challenge for relativistic atomistic first principles calculations), we consider a computationally more efficient approach, where commensurability is assumed, and different *local* stacking configurations are used to model the Moiré pattern, as described next.

The Ir stacking follows ABCABC sequence as shown in Figure 1a. For all the configurations considered, the Ir(111) surface is terminated on A site, and Co atoms (small empty circles in Figure 1) are placed at B position, following a pseudomorphic structure. To model several absorption configurations, carbon, boron, nitrogen and iron atoms are placed at different positions that are labeled following the Ir lattice ordering. Thus, three possible absorption sites for graphene (${\mathrm{Gr}}_{AB}$, ${\mathrm{Gr}}_{AC}$, and ${\mathrm{Gr}}_{BC}$) and three for iron (${\mathrm{Fe}}_{A}$, ${\mathrm{Fe}}_{B}$, ${\mathrm{Fe}}_{C}$) are taken, giving a total of nine possible geometric configurations. An example is shown in Figure 1a, where the two carbon atoms are on Ir A and B positions and the Fe atoms are on A. We denote this as ${\mathrm{Gr}}_{AB}{\mathrm{Fe}}_{A}$ configuration. Black triangles, circles and squares in Figure 1b show the three adsorption sites (A, B, C respectively) considered for Fe on ${\mathrm{Gr}}_{AB}$. Keep in mind, however, that in the real Moiré structure, there will be many other intermediate local arrangement, but we take these simplified configurations as representative of the local properties. For n=2,3, we fix graphene’s first layer at AB sites, and follow the standard AB stacking for graphite (hence ${\mathrm{Gr}}_{AB}$${\mathrm{Gr}}_{BC}$ and ${\mathrm{Gr}}_{AB}$${\mathrm{Gr}}_{BC}$${\mathrm{Gr}}_{AB}$). As for *n* = 1, for each one of these configurations, Fe is placed at A, B and C sites. When *h*-BN is taken as spacer, we perform calculations with only one ML because the magnetic coupling is observed to decay quickly when the number of layers increases in graphene. Notice that in this case, the two different chemical species (B and N) make the two sites inequivalent. Consequently, ${\mathrm{BN}}_{AB}$ is different to ${\mathrm{BN}}_{BA}$, and three additional configurations have to be considered.

All structures are optimized using the conjugate gradient (CG) method at scalar–relativistic level until the forces between atoms are less than 0.01 eV/Å. The geometric optimisations are undertaken allowing all the atoms to move freely out–of–plane, constraining the in–plane geometry to the Ir lattice constant of 2.715 Å. One important aspect during the optimisation procedure is that it is performed from the initial atomic AFM arrangements. From this, a subsequent relaxation follows to study the FM configuration. The strength of the interlayer magnetic coupling, J, is obtained from the difference in energy between FM and AFM configurations, so that a positive J favors AFM coupling.

The growth of the Fe ML is not homogeneous, and a tendency for island formation was observed [7], pointing to preferred absorption site locations. We define the Fe adsorption energies, $E_{ads}$, from the difference between the total energy for Ir(111)/Co/*spacer*/Fe and the energy of the constituents, that is Ir(111)/Co/*spacer* and Fe ML, where *spacer* is either *n*Gr, or *h*-BN:
$$E_{ads}=-\left(E_{T}-E_{Fe}-E_{Ir/Co/n\mathrm{Gr}(h\mathrm{BN})}\right),$$
We have also calculated the magnetic anisotropy energy (MAE) using a recent implementation of the off–site Spin–Orbit coupling (SOC) [26,27,28] in the SIESTA code [21]. This approximation takes into account not only the local SOC contributions to the total energy but also the neighboring interactions between atoms to obtain the total self–consistent energy. As usual, the MAE is defined as the difference in the total self–consistent energy between hard and easy magnetization directions.

## 3. Ir(111)/Co/*n*Gr/Fe

### 3.1. Structural Optimisations

The adsorption energies, $E_{ads}$, magnetic exchange interactions, $J_{Co-Fe}$, and main structural parameters for the different Ir(111)/Co/*n*Gr/Fe [*n* = 1,2,3] geometries studied in this work are presented in Table 1.

As usual, the higher the adsorption energy, the more favorable the structure. Consequently, for *n* = 1, the most stable configurations obtained are ${\mathrm{Gr}}_{AB}$${\mathrm{Fe}}_{\left(A,B\right)}$, ${\mathrm{Gr}}_{AC}$${\mathrm{Fe}}_{\left(A,C\right)}$ and ${\mathrm{Gr}}_{BC}$${\mathrm{Fe}}_{\left(B,C\right)}$. Notice that for all these geometries, there is a Fe atom on top of one of the C atoms in graphene. Consequently, the $z_{C-Fe}$ heights are short (in the range ~2–2.2Å), an indication of a strong interaction between graphene and iron, through hybridisation of Fe 3$d_{z^{2}}$ and C $p_{z}$. Furthermore, for ${\mathrm{Gr}}_{\left(AB,BC\right)}$, there is a Co atom just bellow one of the C atoms (at the *B* site) and a similar hybridisation between graphene and Co strengthens the electronic interaction between the two ferromagnets. Overall the interlayer distances are in reasonable agreement with previous theoretical works that include van der Waals corrections and report distances between graphene and Co in the range 2.0–3.3Å [19,20]. The rumpling, $\delta_{C-C}$, in the graphene layer can also be significant when both carbon atoms in the cell are on top of Co or below Fe.

The geometrical characteristics mentioned above have a close correlation with the values obtained for the magnetic exchange coupling, $J_{Co-Fe}$, which are larger (hundreds of meV) when there is the possibility for a direct path between the metals and graphene. On the other hand, when one of the transition metals falls in the hollow of the graphene lattice, $J_{Co-Fe}$ is reduced to a few tens of meV. Remarkably, Gr${}_{AC}$Fe${}_{B}$, where the graphene ring is centered around Co and Fe ions, has preferential FM alignment, and rather large Co-Fe distance. Consequently, we can argue that in the Moiré superlattice, where there are all the different local arrangements, there will be a competition between strong AF couplings in most local stacking configurations, and relatively moderate FM couplings in others, resulting on an overall AF coupling between intercalated Co and overlayer Fe, as observed in experiments [7].

For n>1 the iron adsorption energies decrease as compared to the monolayered spacer. In particular, for *n* = 2, where the upper C atoms location is the same as for ${\mathrm{Gr}}_{BC}$ in *n* = 1, E${}_{ads}$ follows similar site-dependency (A≪BC) but its value is reduced by up to ~64%. Similarly, for *n* = 3 the upper graphene layer is ${\mathrm{Gr}}_{AB}$ and the E${}_{ads}$ values are reduced by ~60% from the values obtained for (*n* = 1) ${\mathrm{Gr}}_{AB}$. Remarkably, for n>1 the corrugation of the graphene layers is also significantly reduced, even for the layers that are in contact with the metals. The magnetic coupling for n>1 is also substantially suppressed, and preferentially FM ($J_{Co-Fe}\leq 0$), which hints to an oscillatory dependence with the interlayer separation, similar to what is expected for RKKY interactions with metallic spacers, and not to an AF coupling with long exponential decay, as observed for semimetals. Next we will analyse in detail the role that graphene plays in the coupling between Fe and Co.

### 3.2. Electronic Structure and Projected DOS

In order to understand the origin of the strong coupling obtained for one graphene ML, the details of the electronic levels must be studied. We will focus our discussing on the geometry that gives the strongest magnetic coupling: ${\mathrm{Gr}}_{AB}$${\mathrm{Fe}}_{A}$. Figure 2 shows the projected density of states for those atomic orbitals whose orientation points directly towards the other magnetic layer (3$d_{z^{2}}$ and 3$d_{xz/yz}$ for Fe and Co), for one, two and three graphene MLs used as spacer (top three rows in the figure, respectively). The last row plots the same *d* orbitals when the graphene has been removed from the of ${\mathrm{Gr}}_{AB}$Fe${}_{A}$ configuration, keeping the frozen structure for Fe, Co and the underlaying Ir(111) surface. Left and right columns correspond to AFM and FM alignments between Co and Fe, respectively. By comparing the first and last panels we can directly address the effect of graphene on the electronic levels of both ferromagnets. Important changes are observed for Fe 3*d* states due to the presence of the graphene layer, and we identify the peak at around –0.5 eV (highlighted by the arrow) as responsible for the coupling with Co, as we will discuss later. This peak comes from 3$d_{z^{2}}$ orbitals, that point perpendicular to the layers. Without the graphene acting as spacer (lower panel), the positions of the peaks in Co and Fe are less correlated, and the small coupling favors FM. There are additional contributions to the magnetic interaction coming from 3$d_{xz/yz}$ of both species, but they are not enough to promote high values of $J_{Co-Fe}$. For thicker interlayer distances (larger number of graphene MLs), the hybridization between Fe and Co states is quickly reduced, and the values of $J_{Co-Fe}$ become negligible. Notice that the similarity between Fe’s pDOS for monolayer Gr and the trilayer Gr is due to the same stacking order of the last layer of carbon and Fe. Similarly, for the bilayer, the pDOS on Fe resembles that of the bottom panel where graphene has been removed, because Fe is placed on the center of C’s hexagon, where the interaction with carbon is minimized.

Our calculations show a strong correlation between large values for the magnetic coupling $J_{Co-Fe}$, and the presence of the hybridisation peak at –0.5 eV described above. The role played by the graphene spacer (*n* = 1) is further evidenced by plotting the projected density of states on $p_{z}$(C) orbitals (Figure 3). The top panel shows the pDOS for Fe and Co 3*d* levels in the Gr${}_{AB}$Fe${}_{\left(A,B,C\right)}$ configurations, which have *J* values decreasing in strength from left to right. The green filled curve in the lower panel plots the $p_{z}$(C) states for C atoms placed at the B site, i.e., on top of the underlying Co atom. There is a peak of $p_{z}^{↑}$(C) at ~–1.5 eV that reveals the hybridization between Co and the graphene layer in the three geometries shown. However, we argue that the magnetic coupling between Co and Fe is correlated with the presence of the peak at –0.5 eV. For Gr${}_{AB}$Fe${}_{A}$ and Gr${}_{AB}$Fe${}_{B}$, there is a clear overlap between the peaks with red and blue lines in the top panels, and the green/shaded peak in the lower panel. The larger the overlap, the larger the strength of the magnetic interactions ($J_{Co-Fe}\sim$250, and ~100 meV, respectively). On the contrary, the Gr${}_{AB}$Fe${}_{C}$ geometry does not show such peak, neither on the ferromagnets, nor on graphene, and the magnetic coupling is reduced by one order of magnitude.

It is known that the characteristic electronic structure in graphene can be significantly altered by chemisorption on certain metals [29]. The short C–Co or Fe–C distances (~2 Å) are characteristic of these situations, and in these cases graphene’s bands are no longer that of a semimetal. This is evident from the lower panels in Figure 3 and explains why the spacer acts as a (bad) metal for the magnetic super–exchange coupling, with an oscillatory behaviour for the sign of *J* as a function of thickness. When the number of layers increases to n=3, the central layer becomes less hybridized with the FM metals and partially recovers its semimetalic character (central panel in Figure 4). In addition, the distance between Co and Fe is so large (>6 Å) that the hybridized graphene layers close to the metals cannot propagate further the magnetic interactions and the coupling between the magnets becomes negligible. We also find that the buckling in the graphene layer ($\delta_{C-C}$ in Table 1) seems to correlated with the degree of hybridization between $p_{z}$(C) orbitals and the nearby ferromagnets, so that largest $\delta_{C-C}$ correspond to stronger hybridization and stronger magnetic coupling between Fe and Co.

### 3.3. Magnetic Anisotropy Energies

We address now the magnetic anisotropy energy (MAE) for the structures in which graphene layers act as spacer between Fe and Co MLs. We have studied the electronic configurations where the total magnetization is in–plane and out–of–plane, for both FM and AFM alignments between Co and Fe, and the values for the different Fe occupation sites are depicted in Figure 5. The symbols inside the black rectangles correspond to FM configurations whilst the others are for AFM alignments. The positive MAE obtained for all the studied configurations is associated to an easy axis that is out–of–plane. Although there is a significant dispersion in the values reported, we can see that the average perpendicular anisotropy is remarkable, and seems to be in agreement with the experimental measurements [7]. We also observe certain correlation between $J_{Co-Fe}$ (see Table 1) and MAE, with larger values of the magnetic coupling corresponding to larger values of MAEs.

We observe that for spacers with more than one ML (n>1) all the MAEs are stabilized at around 2.5 meV, independently of the stacking configuration. A possible explanation for this could be that the Fe and Co MLs become decoupled (longer interatomic distances, and lower overlap of the electronic levels through graphene’s $p_{z}$ orbitals), and the MAE obtained is that of the isolated ferromagnet’s layers. We can easily check this by computing separately the MAE for a freestanding Fe ML, and a Ir/Co/Gr${}_{AB}$ slab. The obtained MAEs are 0.4 meV and 2.4 meV respectively, which means that the total MAE for the combined multilayered heterostructures is clearly dominated by the strong PMA of intercalated Co, which had been reported before [8].

The relatively small MAE for the freestanding iron ML opens the possibility of having a different easy axis in each magnetic layer: cobalt is clearly out–of–plane (large PMA), but iron could have an in–plane alignment. We explored such possibility, by computing non–collinear arrangements between the magnetic moments in Fe and the perpendicular magnetization in Co. The obtained energies are ~70 meV higher than the counterpart perpendicular configuration, with the AFM alignment remaining as the most stable magnetic configuration.

## 4. Ir(111)/Co/*h*-BN/Fe

We move on now to discuss the properties of the magnetic heterostructure when insulating *h*-BN is used as spacer. Based on the previous results for graphene, where the magnetic coupling is substantially suppressed when the number of layers increases, here we will only consider ML *h*-BN.

### 4.1. Structural Optimisations

Although *h*–BN is isostructural to graphene, the two atomic sites are now inequivalent, which translates into six different stacking configurations in our simplified model geometry. We denote them as BN${}_{\left(AB,AC,BC\right)}$ and NB${}_{\left(AB,AC,BC\right)}$ (See Table 2). The labelling is chosen such that the first specie occupies the site corresponding to the first sub–capital letter and the second specie to the second. For example, for BN${}_{AB}$, B is at the *A* site and N is in *B* site, while for NB${}_{AB}$ N is at *A* and B is at *B* site. For each one of these six stacking, Fe atoms are laid at similar adsorption sites than for graphene, that is Fe${}_{A}$, Fe${}_{B}$, or Fe${}_{C}$.

The highest adsorption energy values, reported in Table 2, are highlighted with an asterisk, and correspond to configurations where the N atom sits just below Fe (BN${}_{AC}$Fe${}_{C}$, BN${}_{BC}$Fe${}_{C}$, NB${}_{AB}$Fe${}_{A}$ and NB${}_{AC}$Fe${}_{A}$). There are two additional geometries that satisfy the same condition: BN${}_{AB}$Fe${}_{B}$ and NB${}_{BC}$Fe${}_{B}$. However, their E${}_{ads}$ is ~60% lower, possibly because in this case the N atom is also strongly hybridized with the underlying Co site (notice the shorter z${}_{BN-Co}$ distances) and this reduces the strength of the chemical bond with Fe (slightly longer z${}_{BN-Fe}$ distances).

As opposed to what happens when graphene is used as spacer, where most of the stacking configurations studied favor AFM coupling, with *h*-BN there is a competition between AFM and FM interactions. We observe that stacks with short z${}_{BN-Fe}$ distances (~2.3Å) tend to favor AFM coupling, while longer distances (~2.9Å) result in FM coupling. As for the case of graphene, the strength of the coupling seems to be correlated with the strength of the hybridization between orbitals, and this can be related to the corrugation (buckling) in the *h*-BN layer. Table 2 shows the values for $\delta_{B-N}$, defined as the difference between z${}_{B}$ and z${}_{N}$. Stronger magnetic couplings happen when $|\delta_{B-N}|\sim$0.22Å.

### 4.2. Electronic Structure and Projected DOS

As we saw in Section 3.2, when graphene is the spacer between Fe and Co MLs there is a direct effect of $p_{z}$(C) orbitals and their hybridisation with 3*d*(Fe/Co) orbitals favors AFM coupling. Now, when *h*-BN is the spacer, approximately half of the structures studied favors FM couplings and the rest show AFM (See Table 2). All structures where Fe sits on top of a N atom have short z${}_{BN-Fe}$ distances and AFM coupling ($J_{Co-Fe}>$0), but there are also geometries with short z${}_{BN-Fe}$ distances and FM coupling (such as NB${}_{AC}$Fe${}_{C}$). To get a better understanding of the origin of the FM/AFM coupling, we analyse next the details of the electronic structure for two representative configurations: NB${}_{AB}$Fe${}_{B}$, which has a strong FM interaction ($J_{Co-Fe}=-126.5$ meV), and NB${}_{BC}$Fe${}_{B}$, which favors AFM coupling ($J_{Co-Fe}=101.2$ meV), Figure 6 and Figure 7, respectively.

Upper panels in Figure 6 plot the pDOS of 3$d_{z^{2}/xz/yz}$ orbitals for Fe and Co (red and blue lines, respectively) in the AFM (left) and FM configurations (right) for NB${}_{AB}$Fe${}_{B}$ stacking. The lower panels show the partial DOS projected on $p_{z}$ orbitals for B and N (green and black lines). First thing to notice is the absence of a gap in *h*-BN, which becomes metallic, in agreement with previous studies of *h*-BN in contact with transition metals [11]. It can be seen that for the AFM configuration there is no significant interaction between *d* orbitals in Fe and Co, since their bands are localized at different energies. Furthermore, there is some hybridization between nitrogen $p_{z}$ orbitals and iron 3*d* (peak at –2.7 eV), but no direct bonding with boron, in spite of the fact that both are at a *B* site. However, in the FM alignment, there is some overlap in the range –2 eV and –3 eV between $d^{↑}$ states of Fe and Co, which is also partially supported by $p_{z}^{↑}$ orbitals coming from both B and N. This, clearly stabilizes the energy and results in a favorable FM coupling.

On the other hand, Figure 7 shows the equivalent pDOS of *d* and $p_{z}$ orbitals for NB${}_{BC}$Fe${}_{B}$ stack, where AFM coupling dominates ($J_{Co-Fe}$ = 101.2 meV). In this case, the N atom is placed just below Fe. The interaction between 3$d_{z^{2}}^{↑}$ orbitals coming from Fe and Co is clearly visible from –2.1 eV up to –1.5 eV, both for AFM and FM configurations. However, mediation through $p_{z}$(N) orbitals only happens for the AFM alignment ($p_{z}$(N) is almost flat in that energy range for the FM case), which stabilizes the AFM over FM.

The above discussion seems to sustain the hypothesis that strong hybridization between $p_{z}$ orbitals in BN and 3*d* orbitals in the transition metals favors short interlayer distances, and AFM couplings. However, when the hybridization is weaker, the longer interlayer distances can favor a FM coupling. We can then expect that sub–monolayer coverings, where small islands of Fe are deposited mostly over BN in regions where the absorption energy is larger, would have stronger AFM couplings, which would decrease due to competition with FM interactions as other regions (with weaker hybridization) are covered.

### 4.3. Magnetic Anisotropy Energies

Similar to what happened for graphene-based heterostructures, *h*-BN seems to favor PMA (the easy axis lays always out–of–plane). The MAE values computed for all the Ir(111)/Co/*h*–BN/Fe configurations studied here are summarized in Figure 8. The dispersion in the MAE ranges from 1.5 to 4.5 meV and it is quite similar for both AFM and FM alignments. Inspecting the data in Table 2 we observe that FM configurations ($J_{Co-Fe}<$ 0) are frequently associated to higher values of the MAEs, as in NB${}_{AB}$Fe${}_{B}$, NB${}_{AB}$Fe${}_{C}$, and BN${}_{BC}$Fe${}_{A}$.

We computed the MAEs for a single Co layer intercalated under *h*-BN with different possible stackins, which are roughly ~2 meV lower than for the corresponding heterostructures with Fe on top. This is in contrast with what is obtained for graphene, where the strong anisotropy of the intercalated Co dominates the total MAE for the heterostructure, as discussed in Section 3.3. However, the strong PMA persists with *h*-BN, which hints to a general enhancement of PMA with ML coatings [8].

## 5. Conclusions

We have carried out a theoretical study of the geometric, electronic and magnetic structures of Ir(111)/Co/*n*Gr/Fe [*n* = 1, 2, 3] and Ir(111)/Co/*h*–BN/Fe heterostructures. Due to the mismatch between graphene and Ir, the minimum size of the unit cell of these ultra-thin films would require at least ~600 atoms. Instead, we have taken a simplified model structure where a commensurable minimal unit cell is considered, and explored the properties of a selected subset of the possible vertical stacking geometries for Gr/*h*-BN over an intercalated Co ML that follows the epitaxial structure of Ir(111). Different adsorption sites of Fe atoms over the honeycomb lattice for graphene/*h*-BN have been considered (See Section 2). In total, we have analysed fourteen different graphene-based configurations and another eighteen configurations using *h*-BN. We have performed structural optimisations for all these geometries, using a CG method at the scalar–relativistic level for both parallel (FM) and antiparallel (AFM) alignments between Fe and Co magnetizations.

The two different spacers between the magnetic films were taken in order to search for different type of magnetic couplings, coming from semimetallic (graphene) or insulating (*h*-BN) materials. However, in the ultrathin limit (one ML), the hybridization between ferromagnets and the spacer heavily distorts the electronic properties of the latter, and both spacers behave as spin–polarized metals. We explored the effect of thicker spacers with bilayer and trilayer graphene, and saw a oscillatory behaviour (antiferro/ferro) and a fast decay in the strength of the magnetic coupling, as expected for metallic spacers in a RKKY model. In terms of designing new synthetic magnetic heterostructures based on 2D materials, we conclude that only ML spacers provide strong magnetic couplings, of hundreds of meV.

We have identified that the origin of the strong coupling between Fe and Co MLs comes from the hybridization of $p_{z}$ orbitals in C/*h*-BN with 3*d* states in the transition metals. In graphene this hybridization is pinpointed by a peak in the density of states at around –0.5 eV and gives AFM coupling, which is stronger the stronger the bonding between the atoms is, which also implies shorter interlayer distances (z${}_{C-Fe}$ and z${}_{Co-C}$ of ~2.1Å), and stronger rumpling in the graphene layer ($\delta_{C-C}\sim$ 0.2 Å). The stronger bonds between graphene and the Fe film also means that the absorption energies are larger, so that sub–monolayer Fe coverage will possibly happen through islands that will have strong AFM coupling to the cobalt substrate.

The situation is a little bit different when the spacer is made of monolayered *h*-BN. Here, strong hybridization between 3*d* orbitals in the transition metals, through the $p_{z}$ orbitals in *h*-BN, also favors short interlayer distances and strong AFM coupling. The hybridization is mainly characterized by a peak in the density of states at around –2 eV. The growth of the Fe film is also likely to happen through islands formation over regions that favor such coupling. However, larger Fe domains will also include regions with different stacking orders, where the absorption energy is less favorable, the interlayer distances (mostly z${}_{Fe-BN}$) are longer, and the magnetic coupling becomes FM. Consequently, we foresee a competition between strong FM and strong AFM domains, and the formation of a complex magnetization texture, which might be appealing for spintronics applications.

For all the configurations considered, we obtain a PMA (MAE positive and in the range of ~3 meV). It is mainly dominated by the strong contribution from the underlying cobalt ML, which is enhanced by the presence of the graphene/*h*-BN on top. For thicker spacing material, the perpendicular anisotropy of the heterostructure is mostly conserved, due to the contribution from the cobalt layer, but the weak coupling with iron means that the orientation of the magnetization in the latter can be tuned by moderate magnetic fields (the MAE for an isolated Fe ML is much smaller), and non–collinear alignments between Fe and Co layers could be obtained.

Although our calculations are based on a very simplified model based on commensurate lattices and not the real Moiré pattern, we believe that our simulations can provide some general guidelines for the design of synthetic magnetic heterostructures based on bidimensional materials, and give some clues on the expected properties that these could have. We expect that the final properties of the Moiré superstructure will be an averaging of the local properties here obtained, which are robust on well defined regions that will have similar stacking orderings as the one considered in the model. The agreement with the available experimental results obtained for Ir(111)/Co/Gr/Fe is remarkable, and points in this direction.

## Figures and Tables

**Figure 1 nanomaterials-09-01764-f001:**
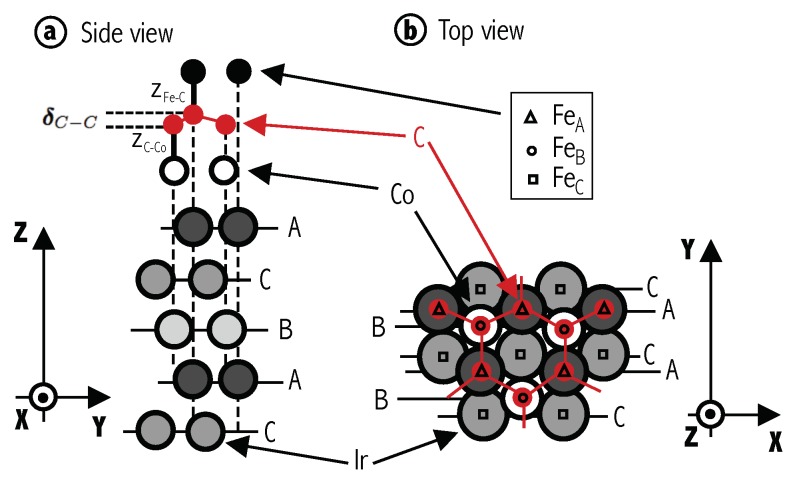
(Color online) (**a**) Schematic side view of a typical stacking configuration for Ir(111)/Co/Gr/Fe heterostructure (in particular, the ${\mathrm{Gr}}_{AB}$${\mathrm{Fe}}_{A}$ configuration). The Ir(111) stacking follows ⋯ABCABC⋯ and is terminated in A. Intercalated Co between graphene and Ir is depicted by empty circles and Fe atoms are represented by black solid bullets on top. Vertical dashed black lines are guides for the eye. Graphene’s corrugation after structural optimization, $\delta_{C-C}$, and the inter-layer distances with Fe, z${}_{C-Fe}$, and Co, z${}_{Co-C}$, are also sketched. (**b**) Top view of Gr${}_{AB}$/Co/Ir structure highlighting the three possible Fe adsorptions sites, A, B and C, as triangles, circles and squares, respectively. Horizontal thin solid lines are guides for the eye.

**Figure 2 nanomaterials-09-01764-f002:**
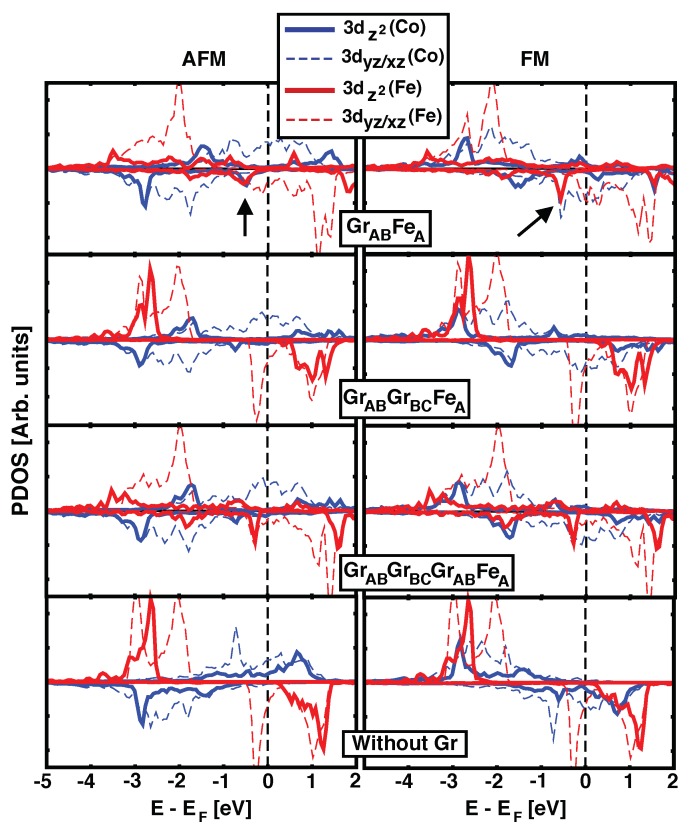
(Color online) Density of states (DOS) projected onto Fe and Co *d* atomic orbitals for AFM (left) and FM (right) configurations in Gr${}_{AB}$Fe${}_{A}$, Gr${}_{AB}$Gr${}_{BC}$Fe${}_{A}$ and Gr${}_{AB}$Gr${}_{BC}$Gr${}_{AB}$Fe${}_{A}$ structures, top three rows. Bottom panel corresponds to the projected DOS for Fe and Co in the Gr${}_{AB}$Fe${}_{A}$ structure in absence of the graphene layer. Thick solid colored lines show 3$d_{z^{2}}$(Fe/Co) orbitals whilst light dashed ones depict $d_{xz/yz}$(Fe/Co). Black arrows highlight the position of the characteristic peak discussed in the text.

**Figure 3 nanomaterials-09-01764-f003:**
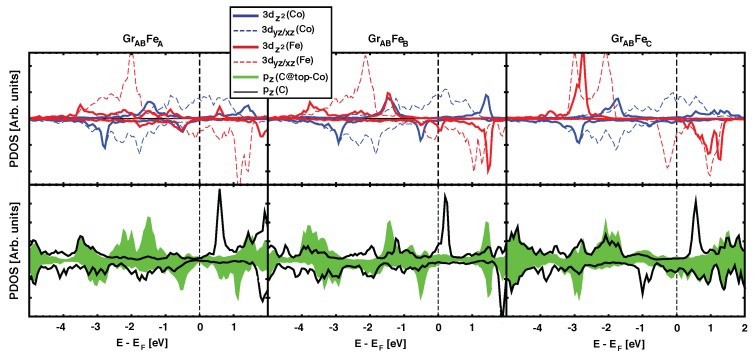
(Color online) Partial DOS for the optimized Gr${}_{AB}$Fe${}_{\left(A,B,C\right)}$ structures in the AFM configuration. Upper panel follows the same color code as in Figure 2. Bottom panel shows the pDOS for $p_{z}$ orbitals in C atoms on *A* and *B* sites (over Co atom) as black and green/filled curves, respectively.

**Figure 4 nanomaterials-09-01764-f004:**
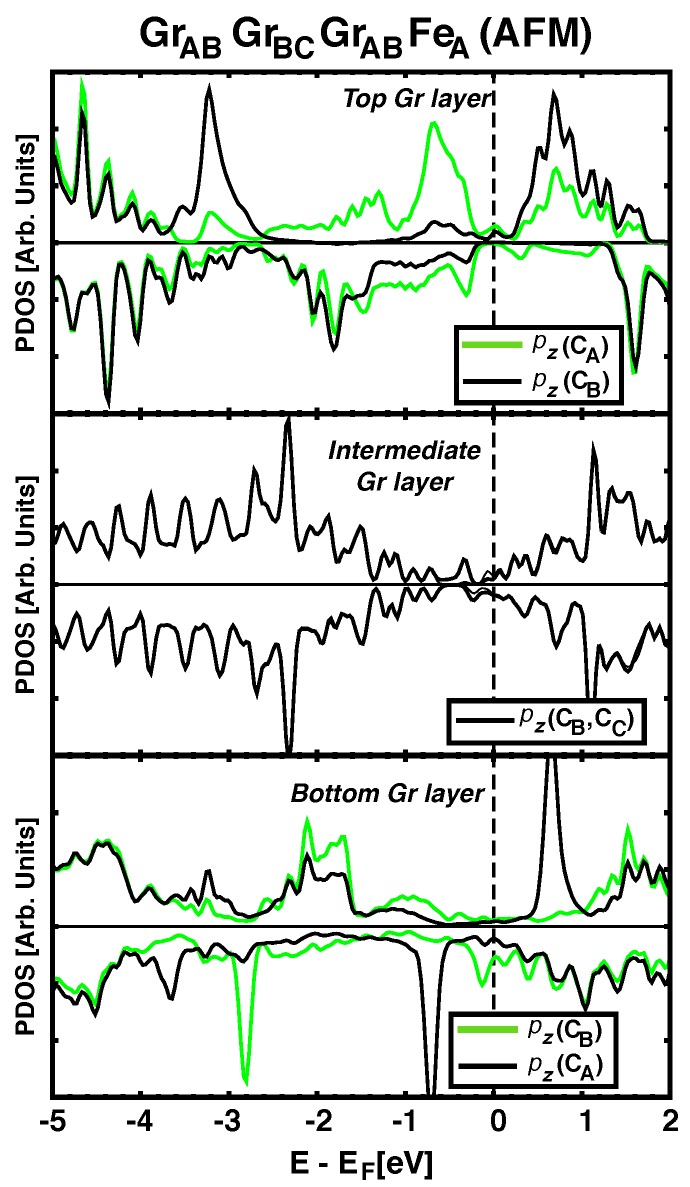
(Color online) Upper, intermediate and lower panels show the pDOS for $p_{z}$ orbitals in each carbon atom of Ir/Co/Gr${}_{AB}$Gr${}_{BC}$Gr${}_{AB}$Fe${}_{A}$ heterostructure. Green solid line depict the pDOS of C atoms just bellow Fe atoms (top row) or just above Co atoms (bottom row). Middle panel shows the pDOS for the intermediate graphene ML, which resembles that of a graphene layer, except for a small spin polarization.

**Figure 5 nanomaterials-09-01764-f005:**
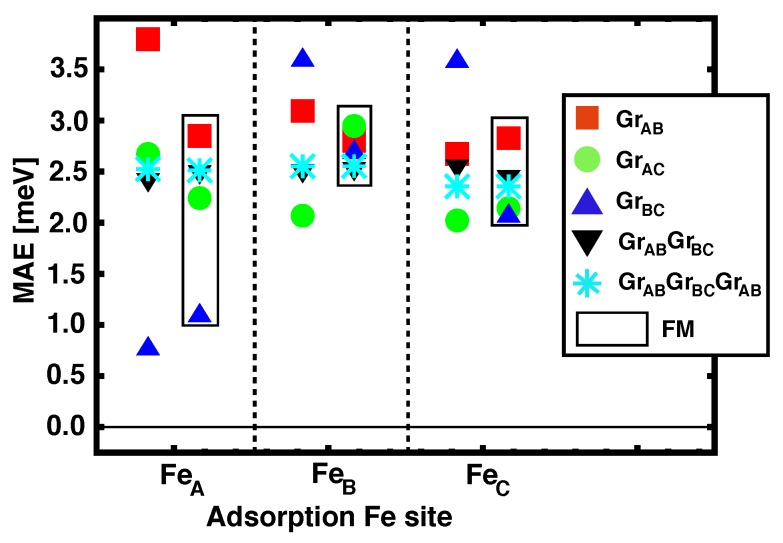
(Color online) Magnetic anisotropy energies for Fe/*n*Gr/Co/Ir(111) [*n* = 1, 2, 3], with different adsorption sites for Fe on graphene. The values obtained for the FM alignment are presented inside the black vertical rectangles.

**Figure 6 nanomaterials-09-01764-f006:**
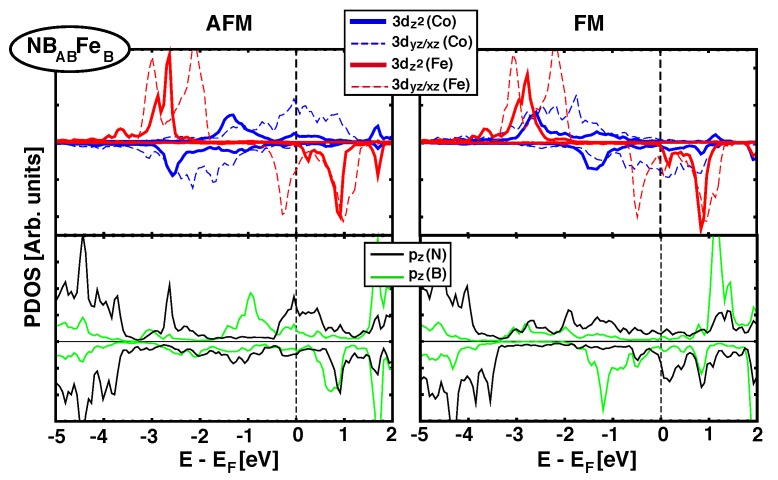
(Color online) Projected Density of states over Fe and Co 3$d_{z^{2}/xz/yz}$ orbitals (top panel) and $p_{z}$(B/N) orbitals (lower panel)for NB${}_{AB}$Fe${}_{B}$ structure. AFM and FM configurations are shown in left and right columns, respectively. The magnetic exchange coupling $J_{Co-Fe}$ is –126.5 meV and favors FM alignment.

**Figure 7 nanomaterials-09-01764-f007:**
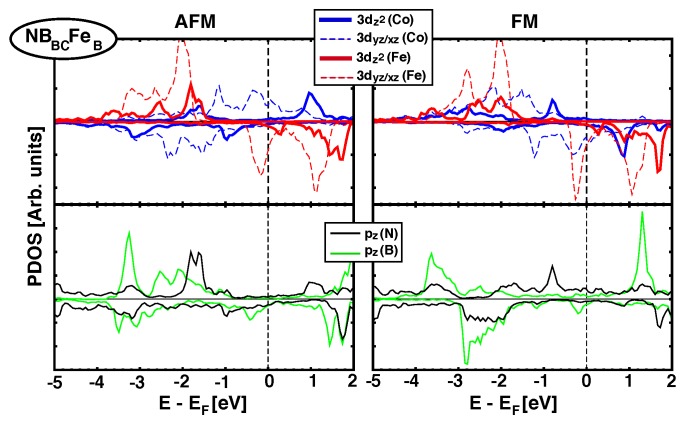
(Color online) Projected Density of states over Fe and Co 3$d_{z^{2}/xz/yz}$ orbitals (top panel) and $p_{z}$(B/N) orbitals (lower panel)for NB${}_{BC}$Fe${}_{B}$ structure. AFM and FM configurations are shown in left and right columns, respectively.The magnetic exchange coupling $J_{Co-Fe}$ is 101.2 meV, and favors AFM alignment.

**Figure 8 nanomaterials-09-01764-f008:**
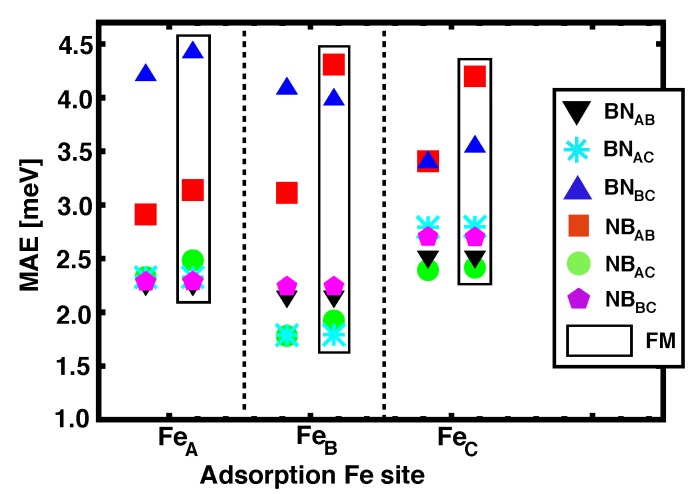
(Color online) Magnetic anisotropy energies values of Ir(111)/Co/*h*–BN/Fe as a function of the adsorption site for Fe on *h*–BN. The values reported inside the black rectangles correspond to the FM alignment between Fe and Co.

**Table 1 nanomaterials-09-01764-t001:** Adsorption energies, $E_{ads}$, magnetic exchange interaction defined as the difference between the FM and AFM total Kohn-Sham energies, $J_{Co-Fe}=E_{FM}-E_{AFM}$, corrugation between C atoms in the first ML, $\delta_{C-C}$, out–of–plane distances, $z_{C-Fe}$ and $z_{Co-C}$, between Fe ML and graphene, and graphene and Co ML, respectively. E${}_{ads}$ is in eV, $J_{Co-Fe}$ in meV and heights in Å. The values in parenthesis correspond to the ones obtained for FM alignment between Fe and Co.

*n*Gr	C Sites	Fe Sites	E_*ads*_ (AFM/FM)	*J* _*Co*−*Fe*_		*δ* _*C*−*C*_	*z* _*C*−*Fe*_	*z* _*Co*−*C*_
1	Gr_*AB*_	A	1.66 (1.41)	254.13		0.27	2.02	2.11
		B	1.61 (1.51)	97.74		0.28	2.21	2.09
		C	0.34 (0.32)	14.13		0.15	2.98	2.06
	Gr_*AC*_	A	1.13 (1.11)	15.74		0.01	2.03	3.10
		B	0.26 (0.27)	−8.47		0.00	2.97	3.14
		C	1.13 (1.12)	15.69		0.01	2.04	3.10
	Gr_*BC*_	A	0.32 (0.31)	10.33		0.01	3.02	2.03
		B	1.41 (1.30)	105.47		0.26	2.21	2.08
		C	1.53 (1.30)	226.63		0.26	2.04	2.10
2	Gr_*AB*_Gr_*BC*_	A	0.20	−0.10		0.01	3.17	2.06
		B	0.97	−3.60		0.01	2.03	2.06
		C	0.99	−1.65		0.01	2.03	2.06
3	Gr_*AB*_Gr_*BC*_Gr_*AB*_	A	0.98	0.01		0.01	2.03	2.06
		B	0.96	−0.07		0.01	2.03	2.06
		C	0.20	0.04		0.01	3.18	2.06

**Table 2 nanomaterials-09-01764-t002:** Adsorption energy, E_*ads*_; magnetic exchange interaction $J_{Co-Fe}$ defined as the difference between the FM and AFM total energies, $E_{FM}-E_{AFM}$; corrugation in the BN layer taken as the difference between B and N heights, $\delta_{B-N}$; out–of–plane distances, $z_{Fe-(BN)}$ and $z_{(BN)-Co}$, between Fe atoms and (BN) and between (BN) and Co, respectively. E_*ads*_ in eV, $J_{Co-Fe}$ in meV, and heights in Å. The values in parenthesis correspond to the FM configuration.

B/N Sites	Fe Sites	E_*ads*_ (AFM/FM)	*J* _*Co*−*Fe*_		*δ* _*B*−*N*_	*z* _*Fe*−*BN*_	*z* _*BN*−*Co*_
BN_*AB*_	A	0.38(0.38)	3.0		−0.15	3.05	2.08
	B	0.92 (0.81)	108.3		−0.22	2.31	2.11
	C	0.44 (0.43)	13.1		−0.14	3.01	2.09
BN_*AC*_	A	0.75 (0.75)	−1.8		0.12	2.19	2.89
	B	0.40 (0.40)	−4.1		0.00	3.10	3.23
	*C	1.48 (1.46)	20.2		0.13	2.20	3.00
BN_*BC*_	A	0.48 (0.52)	−35.1		−0.08	2.91	2.26
	B	0.68(0.76)	−75.4		−0.10	2.94	2.25
	*C	1.65(1.62)	36.8		0.16	2.18	2.87
NB_*AB*_	*A	1.58(1.55)	27.0		0.14	2.19	2.97
	B	0.47 (0.59)	−126.5		−0.12	2.87	2.28
	C	0.48 (0.51)	−34.9		−0.09	2.90	2.31
NB_*AC*_	*A	1.45 (1.44)	12.7		0.13	2.20	3.02
	B	0.41 (0.42)	−6.5		0.00	3.04	3.12
	C	0.76 (0.77)	−7.1		0.12	2.19	2.87
NB_*BC*_	A	0.44(0.43)	11.2		−0.14	3.05	2.08
	B	0.92(0.81)	101.2		−0.22	2.33	2.11
	C	0.85(0.77)	83.8		0.23	2.17	2.11

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
