# Peer review of "Guidelines for Selecting Interlayer Spacers in Synthetic 2D-Based Antiferromagnets from First-Principles Simulations"

_nanomaterials, 2019, doi:10.3390/nano9121764_

Round 1
Reviewer 1 Report
The manuscript presents a computational study of synthetic 2D-based antiferromagnets using ab initio DFT-based calculations. The topic is interesting and timely as it tries to shed some light on recently published experimental experiments by giving a microscopic interpretation of the results found in these publications. The paper is, in general, well written. The authors use state-of-the-art computational techniques used in its development and their scientific discussion is clear and free of evident methodological errors. For this reason I suggest to accept the publication of the paper although I would suggest some minor modifications to improve its legibility.
1) The authors should explicitly describe the computational model that they use. First of all they should discuss about the choice of support. Why iridium? I suppose that this is because they use the same case as in experiments in some of the references, but this should be discussed here. Is the choice of the support important or any metal with reasonable matching lattice dimensions is fine?Although they already give a description of the geometry of the system, I feel that in some places some details should be added or explained with more clarity. For instance, how many Ir layers are there in the model? I suppose that the five layers shown in figure 1, but this should be clearly stated in the text. The enplane geometry is fixed during the optimization by fixing the lattice constant, but are all atoms, including Ir, free to move in the perpendicular direction? I suppose the authors use periodic copies of the slab in the perpendicular direction. What is the separation between slabs?I understand that they use a minimal model with on Fe and one Co per repeat unit. This should also be explicitly stated in the text.
2) The analysis of the DOS is somewhat obscure. It is very difficult to see anything in these very crowded DOS plots. Maybe it would be easier to have simpler plots with the z2 and the xz, yz components are plotted in separate panels to avoid graphs with four curves that are difficult to interpret?
When looking at figure 2 it is evident that the z2 component of Fe is very different in the case with one and three Gr layers from the case with two or no Gr layers (see for instance the positive peak at approx. -3 eV). Why is this so? Could the authors find a plausible explanation and see if this is relevant?
3) There is a small problem with two references for which no number appears (lines 136+1 and 236)
Author Response
Dear Editor
Subject: Manuscript nanomaterials-664315
We acknowledge the review of the manuscript by the referee and we are grateful for the constructive comments, which we believe help to improve the quality of the manuscript. The complete list of changes made in the revised manuscript, which we are resubmitting, is appended after our responses to the referee (we use bold face when quoting the referee):
Sincerely yours,
R. Cuadrado and M. Pruneda _____________________________________________________________________
Response to Reviewer # 1 is given below:
1) The authors should explicitly describe the computational model that they use. First of all they should discuss about the choice of support. Why iridium? I suppose that this is because they use the same case as in experiments in some of the references, but this should be discussed here. Is the choice of the support important or any metal with reasonable matching lattice dimensions is fine? Although they already give a description of the geometry of the system, I feel that in some places some details should be added or explained with more clarity. For instance, how many Ir layers are there in the model? I suppose that the five layers shown in figure 1, but this should be clearly stated in the text. The enplane geometry is fixed during the optimization by fixing the lattice constant, but are all atoms, including Ir, free to move in the perpendicular direction? I suppose the authors use periodic copies of the slab in the perpendicular direction. What is the separation between slabs?I understand that they use a minimal model with on Fe and one Co per repeat unit. This should also be explicitly stated in the text.
The main reason why we use Ir(111) as substrate is that graphene grows with high structural quality on this material whilst graphene growth on Pt(111) or Ru(0001), for example, displays a variety of orientations. This observation was pointed out by Johann Coraux, et al (Nanoletters 8, 565 (2008)), and was used to intercalate Co under graphene by Decker et al (Phys. Rev. B 87, 041403, (2013)), which was later used as starting point for the experiments that measured the strong antiferromagnetic interaction with Fe deposited on top. In this manuscript, we followed a successfully synthesised magnetic heterostructure to analyse the effect of using different interlayer spacers.
We thank the referee for her suggestions, which we agree will improve our manuscript. We have added within the text an improved description of the computational model that we have used (section 2 lines 104, 107 and 111 in red for clarity)
2) The analysis of the DOS is somewhat obscure. It is very difficult to see anything in these very crowded DOS plots. Maybe it would be easier to have simpler plots with the z2 and the xz, yz components are plotted in separate panels to avoid graphs with four curves that are difficult to interpret?
This is a hard issue, and we have tried different ways to report the main results obtained from the calculations. We agree with the referee that the plots are difficult to interpret, and require some careful view from the reader. We believe that removing lines, or adding more panels will not facilitate the view of these figures. However, and following the suggestion of the referee we have updated the figures that plot both z^2
and xz/yz pDOS and we have replaced xz/yz solid lines by dashed thin lines, giving more value to z^2 pDOS curves (Figures 2, 3, 6 and 7). We have modified as well the captions if necessary.
When looking at figure 2 it is evident that the z2 component of Fe is very different in the case with one and three Gr layers from the case with two or no Gr layers (see for instance the positive peak at approx. -3 eV). Why is this so? Could the authors find a plausible explanation and see if this is relevant?
The referee is right in pointing out the differences between these panels. The explanation is simple when the specific atomic arrangement is taken under consideration. The top and third panels in the figure 2 correspond to configurations where Fe is placed on A site, on top of a Gr_AB layer. The direct interaction between Fe is different in this case to the situation with the bilayer, where the last graphene layer is Gr_BC (hence Fe is at the centre of the carbon hexagon), or when there is no graphene at all (bottom layer). We believe that these fine details of the atomic arrangements are critical, and have been discussed throughout the manuscript. However, we thank the referee for giving us the opportunity to further stress this effect, by including a new sentence at the end of the first paragraph of the subsection 3.2 (in red for clarity).
3) There is a small problem with two references for which no number appears (lines 136+1 and 236)
We have corrected the missing reference in the main text. Now lines 141+2 and 248.

Reviewer 2 Report
The manuscript titled “Designing guidelines for synthetic 2D-based antiferromagnets from atomistic first-principles simulations“ by R. Cuadrado and M. Pruneda, is a theoretical study on structural, electronic and magnetic properties of Ir/Co/spacer/Fe interfaces where semi-metallic (graphene) and insulator (hBN) spacers are considered. Moreover, the effect of the thickness of the spacer on the magnetic properties is also considered for the case of graphene considering up to 3 graphene layers. The authors, using density functional theory calculations, including spin-orbit terms on model structures calculate adsorption energies, magnetic exchange interaction and magnetic anisotropy energies and rationalise the origin of the coupling between Fe and Co by inspecting projected densities of space metal d orbitals and the Pz orbitals of the spacers. Besides the simplified model used by the authors, in my opinion, the present research contains interesting results that are very well explained, the manuscript is well written and I suggest the publication in the Nanomaterials journal after few minor points will be addressed. Here below I report some needed clarification and suggestions the author can take into account.
Title: Despite the interesting messages coming from the present work, the title seems to be a bit misleading. The authors study the interaction between two metals considering two spacers and different thickness for one of the case. The result that only ML spacers provide strong magnetic coupling is not surprising and the generic title “Guidelines for synthetic 2D-based antiferromagnets” seems to me too general.
Model: The authors avoid to simulate the Moire superstructure of the interface (10x10 graphene unit cell and 9x9 Co/Ir lattice) and takes into account the Moire by assuming commensurability and performs calculations considering different local stacking configurations, claiming that the simulation of the whole system remains a challenge for atomistic first principle calculations. In my opinion the used model is sound, nevertheless, the entire Moire pattern of Ir/Co/graphene has been simulated in the past (see e.g. R. Decker et al, PRB 87, 041403(R) (2013) and G. Avvisati Nano Lett, 18 2268 (2018)). In my opinion, the authors should mention these works and also compare the Co-C distance found for ML graphene with the ones reported in these works. This will allow for instance to evaluate the impact of the Fe layer on the Gr/Co distance and a comparison of the computed distances for the different absorption sites considered by the authors with simple Moire models (see e.g Fig. 2 of Avvisati et al. J. Phys. Chem. C 2017, 121, 1639) would also provide an indication of the occurrence of the chosen configuration in the superlattice.
Methods: The authors perform the DFT calculations by using GGA (PBE) exchange-correlation potentials. Anyway, calculations performed on Ir/Co/Gr interfaces structural relaxations were performed by including van der Waals interactions. (see e.g. R. Decker et al, PRB 87, 041403(R) (2013) and G. Avvisati Nano Lett, 18 2268 (2018)) The authors should justify why they do neglect such interactions and what should they expect by the inclusion of this term on theory relaxed structures.
Missing references: page 4 (line 136); page 8 (line 236)
Typo: page 12, line 329 and 356 later —> latter
Author Response
Dear Editor
Subject: Manuscript nanomaterials-664315
We acknowledge the review of the manuscript by the referee and we are grateful for the constructive comments, which we believe help to improve the quality of the manuscript. The complete list of changes made in the revised manuscript, which we are resubmitting, is appended after our responses to the referee (we use bold face when quoting the referee):
Sincerely yours,
R. Cuadrado and M. Pruneda _____________________________________________________________________
Response to Reviewer # 2 is given below:
1) Title: Despite the interesting messages coming from the present work, the title seems to be a bit misleading. The authors study the interaction between two metals considering two spacers and different thickness for one of the case. The result that only ML spacers provide strong magnetic coupling is not surprising and the generic title “Guidelines for synthetic 2D-based antiferromagnets” seems to me too general.
Although we have not explored different magnetic metals, we believe that our manuscript provides guidelines for selecting appropriate interlayer spacers based on the typical parameters that could play a critical role (metallic vs insulator, and thickness). Furthermore, even if we agree with the referee that it is obvious that “the thinner the better” it is not obvious what would be the interaction screening in a semimetal such as graphene. We have however taken into consideration his/her suggestion and adapted the title to something that we believe might be more appropriate: "Guidelines for selecting interlayer spacers in synthetic 2D-based antiferromagnets from first-principles simulations”
2) Model: The authors avoid to simulate the Moire superstructure of the interface (10x10 graphene unit cell and 9x9 Co/Ir lattice) and takes into account the Moire by assuming commensurability and performs calculations considering different local stacking configurations, claiming that the simulation of the whole system remains a challenge for atomistic first principle calculations. In my opinion the used model is sound, nevertheless, the entire Moire pattern of Ir/Co/ graphene has been simulated in the past (see e.g. R. Decker et al, PRB 87, 041403(R) (2013) and G. Avvisati Nano Lett, 18 2268 (2018)). In my opinion, the authors should mention these works and also compare the Co-C distance found for ML graphene with the ones reported in these works. This will allow for instance to evaluate the impact of the Fe layer on the Gr/Co distance and a comparison of the computed distances for the different absorption sites considered by the authors with simple Moire models (see e.g Fig. 2 of Avvisati et al. J. Phys. Chem. C 2017, 121, 1639) would also provide an indication of the occurrence of the chosen configuration in the superlattice.
We thank the referee for pointing out the soundness of our model. We have added the references that the referee suggested and we have included within the main text a brief comparison with our Co-C distances. Overall the interlayer distances between graphene and Co are in good agreement with previous calculations. This could be due to the strong hybridisation observed in some regions of the Moire pattern, but also to the presence of the iron layer on top. We have added a new comment about this in subsection 3.1 in line 158 (in red for clarity).
3) Methods: The authors perform the DFT calculations by using GGA (PBE) exchange-correlation potentials. Anyway, calculations performed on Ir/Co/Gr interfaces structural relaxations were performed by including van der Waals interactions. (see e.g. R. Decker et al, PRB 87, 041403(R) (2013) and G. Avvisati Nano Lett, 18 2268 (2018)) The authors should justify why they do neglect such interactions and what should they expect by the inclusion of this term on theory relaxed structures.
We agree that in these systems one would expect that the van der Waals interactions should be important. However, we have compared our calculated Co-C distances with those reported in G. Avvisati et al, Nano Lett 18, 2268 (2018) and we observe that our calculations present the same dispersion in the values ranging from ~2.0 to ~3.1, similarly to the values obtained by them (see figure 2-b). We argue that this could be due to the fact that in our model we replace the complete Moiré pattern by local geometric configuration and hence the chemical bonds are more robust.
4) Missing references: page 4 (line 136); page 8 (line 236) Typo: page 12, line 329 and 356 later —> latter
We have modified the missing references in lines 136 and 236 being now the lines 141+2 and 248. The typo in page 12 was also corrected.